# Efficacy of Biological and Chemical Control Agents against *Heterobasidion* Spore Infections of Norway Spruce and Scots Pine Stumps on Drained Peatland

**DOI:** 10.3390/jof9030346

**Published:** 2023-03-11

**Authors:** Tuula Piri, Markku Saarinen, Leena Hamberg, Jarkko Hantula, Talis Gaitnieks

**Affiliations:** 1Natural Resources Institute Finland (Luke), Natural Resources, Forest Health and Biodiversity, Latokartanonkaari 9, 00790 Helsinki, Finland; 2Natural Resources Institute Finland (Luke), Natural Resources, Forest Management, Tietotie 4, 31600 Jokioinen, Finland; 3Latvian State Forest Research Insitute Silava, Forest Phytopathology and Mycology, 111 Riga Str., LV-2169 Salaspils, Latvia

**Keywords:** Heterobasidionroot rot, spore infection, stump treatment, urea, *Phlebiopsis gigantea*, peatland

## Abstract

Treatment of conifer stumps with a control agent effectively prevents *Heterobasidion* spore infections in summer cuttings and protects the residual stand and the next tree generation from damage caused by Heterobasidion root rot. Thus far, stump treatment experiments have been carried out in mineral soils, and no information is available on the efficacy of stump treatment agents in boreal peatland conditions. In the present study, biological and chemical control agents (*Phlebiopsis gigantea* and urea, respectively) were tested in Scots pine and Norway spruce stands subjected to thinning, cap cutting, and clearcutting on drained peatland in Central Finland. The control efficacy of urea was high in both spruce and pine stumps (on average 99.5 and 85.3%, respectively), while the efficacy of *P. gigantea* was highly variable on both tree species and ranged from full protection down to negative control effect, i.e., there were more *Heterobasidion* infections on the treated than untreated half of the stumps. The moisture content of the stump wood or the thickness of the peat layer did not affect the control efficacy of either control agent. These results emphasize a need for further studies to determine the reasons for the unsteadiness of the biological control in peatland conditions.

## 1. Introduction

Root and butt rot caused by *Heterobasidion* species is economically one of the most damaging diseases of conifer forests in the Northern Hemisphere [1,2]. Infection of the exposed stump surface by air-borne fungal spores is the most common pathway and, thus, of critical importance to the establishment of Heterobasidion root rot in commercial forests. The pathogen colonizes the stump and its root system and spreads to the roots of adjacent healthy trees via root contacts [3,4]. As a result of this mycelial spread, a single *Heterobasidion* genotype may infect dozens of conifer trees during the rotation time and even infect trees of the next stand generation [5,6]. Once the pathogen has reached the root systems, its further spread is hard to control. Therefore, prophylactic control methods preventing primary spore infections provide the best opportunity to reduce losses caused by Heterobasidion root rot in intensively managed conifer forests.

Currently, there are only two methods available to protect freshly cut conifer stumps from *Heterobasidion* sp. spore infection. The loggings can be restricted to the cold season when the temperature is below zero degrees and the fungus is not sporulating [7], or spore infections can be controlled by stump treatment during the sporulation period [8].

In Finland, both biological and chemical control agents are used to protect conifer stumps from *Heterobasidion* spore infections. The biological control agent contains oidia (asexual spores) and mycelium fragments of *Phlebiopsis gigantea* (Fr.) Jülich. This common saprophytic basidiomycete is widely distributed throughout the conifer forests of the Northern Hemisphere and is a strong competitor of *Heterobasidion* sp. on a fresh stump surface [9]. The commercial product of *P. gigantea*, Rotstop^®^, was formulated in Finland by Verdera Inc. in 1991 [10].

In chemical treatment, the active ingredient is urea. The control effect of urea is based on the increased pH value on the stump surface achieved by the hydrolysis of urea by urease enzyme in the living sapwood tissues, resulting in the formation of ammonia and a rise in pH up to a level where germination of spores and viability of mycelium of *Heterobasidion* spp. are suppressed [11]. Several commercial urea products in concentrations between 32 and 33% (w/v) are available for stump treatment in Finland.

The efficacy of both biological and chemical control agents has been tested in several experiments conducted in a variety of conditions in Finland and elsewhere, e.g., [10,12,13,14,15,16,17,18]. In general, the efficacy of urea and *P. gigantea* has been high, preventing the vast majority of *Heterobasidion* spore infections and reducing the area of fungal colonies [19,20,21,22]. On the other hand, results showing insufficient control effects have also been reported [13,23,24,25]. The ineffectiveness of the stump treatment may be due to several reasons, such as low concentration [12] or the inadequate application of a control agent on the stump surface [13,26,27], or exceptionally high spore load of *Heterobasidion* spp. [28]. Environmental conditions, stump size, and properties of stump wood, including wood moisture content, have also been shown to have a decisive influence on the control result [23,24,25,29,30].

Most field studies on the efficacy of stump treatment agents have been carried out in conifer forests growing on mineral soils. However, in Scotland and northern England, the efficacy of urea was tested on both mineral and peat soils and was found to fail in providing a consistent control on peat soil subjected to high annual rainfall [28]. The failure associated with urea treatment was assumed to be linked to the absence of hydrolysis in the dead heartwood, which—rather than the sapwood—is the target for *H. annosum* in spruce stumps in wetter climates [28,31].

In southern and central Finland, two *Heterobasidion* species, *H. parviporum* Niemelä & Korhonen and *H. annosum* sensu stricto (Fr.) Bref, cause damage to forests. In this area, ca. 2.33 million hectares of forests grow on drained peatland [32]. Until recently, loggings on peatlands have almost solely been conducted in the winter when the ground is frozen, and *Heterobasidion* fruiting bodies do not sporulate, which may have protected these stands from infections. Improved planning tools for logging operations and the development of harvesting and extraction machinery suitable for sensitive peatland sites to minimize soil disturbance have increased summer loggings on drained peatlands [33,34,35]. Therefore, the situation concerning Heterobasidionroot rot may deteriorate rapidly due to changes in logging practices. Recently, both *Heterobasidion* spore infections and mycelial spreading have been observed to be increasing in Norway spruce stands on peat soils in Latvia [36,37]. Moreover, in Finland, especially in the southernmost parts of the country, Heterobasidion root rot has become more common in spruce and pine forests on drained peat soils [38]. However, no studies on the efficacy of stump treatment have been carried out in Nordic peatland conditions.

The aim of the present study was to test whether stump treatment with *P. gigantea* (Rotstop^®^ SC) and 32.5% urea solution (commercial product PS-kantosuoja-2) provided sufficient protection against *Heterobasidion* spore infection in harvested spruce and pine forests on drained peatland in Central Finland.

## 2. Material and Methods

### 2.1. Study Sites

The stump treatment experiments were conducted in five Norway spruce and five Scots pine stands in Central Finland. All experimental areas have a decades-long drainage history, with a post-drainage vegetation succession developed from original *Sphagnum* moss-dominated mire vegetation towards transformed peatland forest vegetation. In the Finnish site type classification system [39], these transformed peatland sites, as well as all those drainage areas where the vegetation is just progressing towards the end stage of succession, can be classified into five site types according to their vegetation, productivity, and nutrition level: Herb-rich type (Rhtkg), *Vaccinium myrtillus* type (Mtkg), *Vaccinium vitis-idaea* type (Ptkg), Dwarf shrub type (Vatkg) and *Cladonia* type (Jätkg). Further, all these site types are divided into two subtypes (subtype I and II) depending on the hydrology of the original mire in its natural state before the drainage. Subtype II origins from either open treeless mires or partly open pine or spruce mires with a combination of open wet parts and dryer hummocks with tree stands (so-called composite pine or spruce mires). Subtypes I, in turn, have originally been drier and entirely forest-covered mires with hummock-like vegetation and significantly lower water table levels.

In our experiments, the spruce stands were either herb-rich or *Vaccinium myrtillus* types (Table 1). They were mainly subtype I (Rhtkg I or Mtkg I), with the exception of the Hämeenlinna experiment representing subtype II (Mtkg II). Pine-dominated experimental stands were all subtype II peatland forests with the nutrient status of *Vaccinium vitis-idaea* (Ptkg II) and partly *Vaccinium myrtillus* (Mtkg II). The risk of nutrient leaching was concretely expressed in part of the Keuruu 1 experiment, where mild potassium deficiency symptoms were observed.

The experimental stands were subjected to different logging methods: clearcutting, cap cutting, and thinning. The area of the clearcuttings varied from two to five hectares, and the area of cap cuttings was 0.1 to 0.3 hectares. The location and main characteristics of the forest stands are given in Table 1.

Depending on the size of the study area, 10 to 20 soil core samples evenly distributed over the area were taken to determine the soil pH value. Before taking the soil core (ca. 15 cm in depth), the green part of mosses and other living surface vegetation was removed. The pH value was measured by dissolving 5 g of dried soil sample into 25 mL of distilled water and calculated as the mean of all the replicates of each site.

### 2.2. Stump Inoculation

The stands were harvested normally by a forest harvester, but the experimental stumps were left ca. 0.8 m in height. The stumps were randomly selected, considering, however, that they were evenly distributed across the logging area, and the thickness of the peat layer in the vicinity of the stumps exceeded 30 cm.

On the same day or the day after harvesting, the stumps were sawn with a chainsaw to a height of ca. 50 cm to expose a fresh stump surface. Wood samples for the measurement of moisture content (c. 2 × 4 cm in size) were taken both from the heartwood and sapwood immediately after lowering the stump and stored in separate grip-seal plastic bags. The stump surface was then sawn evenly and divided into two halves using a marker line. One half of the stump surface was covered with a weatherproof sheet of paper, and the other half was sprayed with the control agent (*P. gigantea* or urea). This was made to eliminate the variation in wood properties (such as wood moisture content) between the treated and non-treated controls. A total of 134 Scots pine stumps were treated with *P. gigantea* and 125 stumps with urea. The corresponding figures for treated Norway spruce stumps were 148 and 143, respectively.

Rotstop^®^ SC solution was prepared according to the manufacturer’s instructions, and urea was a commercial ready-made 32.5% solution (PS-kantosuoja-2, JL-Tuotteet Oy). An approximately 1 mm thick layer of the control agent was sprayed on the uncovered half of a stump surface. About ½–1 h later, both halves were sprayed with a conidial suspension of *Heterobasidion*. The *Heterobasidion* inoculum was prepared on the same day by mixing conidia of four local *Heterobasidion* strains collected from stumps and standing spruce and pine trees on peat soil. To inoculate pine and spruce stumps, strains of *H. annosum* and *H. parviporum*, respectively, were used. Spores were harvested from 3- to 4-week-old fungal cultures growing on malt agar plates at room temperature. The surface was washed to obtain conidia concentration, which was adjusted to ca. 10 to 50 spores cm^−2^. The number of living spores was determined after the treatments by spraying an approximately 1 mm layer of suspension onto five malt agar plates. After four days of incubation at ca. 20 °C, the average number of germinated spores/1 cm^−2^ was calculated (Table 2). Furthermore, the treatment suspensions of *P. gigantea* were prepared on the morning of the same day they were used.

Stumps were labeled after the treatments. Some (6–15) stumps were left as untreated controls in sites where the number of stumps was sufficiently high for this extra control to obtain information on the amount of natural *Heterobasidion* spore load. When experimental stands were located close to each other, only one set of controls was left for the whole area.

The wood samples for moisture content (MC) determinations were kept in the fridge overnight, and the fresh weights were determined the next day. The wood samples were then stored in a freezer for one to two weeks. The dry weight was determined after drying the samples at 70 °C until no further weight loss was observed. MC was determined twice for each stump, i.e., before stump treatment and before harvesting, and calculated by applying the following equation: MC% = 100 × (weight of water in wood/oven-dry weight of wood).

### 2.3. Sampling

Seven pine and spruce stumps had to be discarded because they had been overdriven during timber forwarding, not found, or had old decay caused by *Stereum sanquinolentum* or *Armillaria* sp. The rest of the stumps were sampled 9–12 weeks after treatment by cutting two 3–4 cm-thick discs from the top of each stump. The upper disc was discarded, and the lower sample disc was used in the analysis. The wood samples for the measurement of moisture content were sawn under the lower sample disc and processed as described above. The sample discs were barked, washed, and incubated at room temperature for 5–7 days in polythene bags. After incubation, the area occupied by *Heterobasidion* sp. (based on conidiophore formation) on the upper surface of the disc was determined under a dissection microscope. The outlines of *Heterobasidion* colonies were marked on the surface of the disc and then traced onto a sheet of transparent paper. The area of the fungal colonies, as well as the area of both halves of the discs, was measured using the ImageJ program (public domain). Before measuring, a 1 cm wide strip on both sides of the borderline between the treatment and control was reduced from both halves of the surface area. Because *Heterobasidion* infections had only exceptionally reached the depth of 8–10 cm from the treated surface, the underside of the discs was not analyzed. The percentage area occupied by *P. gigantea* on the disc surface, based on the characteristic orange-brown color on the disc surface (hyphae, double clamps, and oidia), was not measured but was roughly estimated visually.

Additional discs from the stumps with exceptionally large *Heterobasidion* colonies on the disc surface were sawn from ground level and the base of the main roots and analyzed to ensure that decay in the stumps did not originate from infections already established in the root system before the trees were felled. Since there were no *Heterobasidion* infections in the lower part of these stumps, it was concluded that they were infected through the stump surface.

### 2.4. Data Preparation

Sample disc analyses revealed that—despite the artificial infection—38.7% of all inoculated pine stumps and 6.7% of spruce stumps were not infected, i.e., there were no *Heterobasidion* infections in the control half of the stump surface. These stumps were excluded from the statistical analyses. In addition, the clear-cut pine stand in Kuru, where *Heterobasidion* inoculation failed almost completely, was excluded from further analyses. Thus, the final number of analyzed pine and spruce stumps were 151 and 265, respectively.

The control efficacy (E%) of *P. gigantea* and urea was calculated by comparing the area occupied by *Heterobasidion* sp. on both (treated and untreated) halves of the disc, using the following formula:E% = 100 − (100 × nt/nu)(1)
where nt and nu represent percentages of the area occupied by *Heterobasidion* sp. in treated and untreated halves. Only stumps with successful infection by *Heterobasidion* spores on the untreated half of the disc were included in the analyses. For each site, the control efficacy (E%) was calculated as means by tree species and treatments. For tree species by treatments, the control efficiency was calculated from the mean values of each site.

### 2.5. Statistical Analyses

The effects of different treatments on the cover of *Heterobasidion* sp. in different harvesting methods were investigated by using statistical models. Models were estimated separately for spruce and pine stumps. Each treatment area, i.e., the half of the stump treated with *P. gigantea* or urea, and the untreated control half, was considered as a separate observational unit in the data, although the control and the treatments (*P. gigantea* or urea) were applied on the same stumps. Thus, spruce models included 535 and pine data 351 observational units.

The percentage of the area (including sapwood and heartwood) occupied by *Heterobasidion* sp. (%) was a response variable in the models. Logit transformation was applied to cover values when models were estimated [40]. However, 0.1 units were added to each cover value to include investigated stump areas with zero values for analyses. The effects of different treatments (*P. gigantea* or urea) and harvesting methods (clearcutting, cap cutting, or harvesting) on the percentage of area occupied by *Heterobasidion* sp. on stump surfaces were investigated using linear mixed models with function lme in library nlme in the statistical program R [41,42]. Explanatory variables were (1) treatment (as a factor with three levels: control, i.e., no treatment, *P. gigantea* or urea), (2) harvesting method (a factor with three levels: clearcutting, cap cutting or thinning), (3) diameter of a stump (cm, without bark), (4) moisture content of sapwood before the treatments (%), (5) moisture content of heartwood before the treatments (%), (6) peat thickness (cm), and (7) an interaction term between treatment and harvesting method. However, clearcutting was not applied for spruce stands, i.e., it was not included as a harvesting method for spruce models. Furthermore, the thickness of the peat layer could not be included in the pine model since it correlated too strongly with harvesting methods. The treatment year, and identification code for each experimental stand and stump, were included in the models as nested random factors to take into account the fact that different treatments were performed within the same years (2020, 2021, or 2022) and sites and treatment-control combinations within same stumps. Treatment year as a random factor was excluded from the pine model since only one stand was treated in 2021.

Predicted values in different treatments and harvesting methods were calculated based on estimated models using the library *AICcmodavg* [43]. Other variables in the modes (stump diameter, sapwood and heartwood moisture, and peat thickness) were kept in their means when the predicted values were calculated.

Kruskal-Wallis H-test with post hoc tests was used to test differences in the control efficacy (E%) between the *P. gigantea* and urea treatments by tree species. An independent-samples *t*-test was used to compare the means of single variables (MC% and proportion of the cross-sectional area occupied by *Heterobasidion* sp.) between spruce and pine stumps. The analyses were performed using SPSS Statistic Software 15.01 for Windows.

## 3. Results

### 3.1. Artificial and Natural Heterobasidion Infections

The variation in the success of artificial infection was high, especially among the pine stands: The proportion of infected stumps varied from 5.9 to 96.6% in pine stands and from 76.3 to 100% in spruce stands. In addition, the proportion of the area occupied by *Heterobasidion* sp. on the control half of the stump surface was significantly larger on spruce stumps than on pine stumps. i.e., mean 16.7 and 5.4%, respectively (*t* = −10.325, df = 416.775, *p* < 0.001).

Natural infection by *Heterobasidion* spores was completely absent in all other stands except spruce thinning in Suonenjoki, where all four untreated control stumps were infected naturally by *Heterobasidion* spores, and spruce cap cutting in Hämeenlinna, where 11 out of 12 control stumps were naturally infected.

### 3.2. Efficacy of Urea and P. gigantea Treatments

A total of 47.5% of pine stumps treated with *P. gigantea* and 1.4% of pine stumps treated with urea were infected by *Heterobasidion* spores. Correspondingly, in spruce stands, 71.2% of *P. gigantea*-treated and 59.5% of urea-treated stumps were infected by *Heterobasidion* sp.

The control efficacy (E%) within the experimental stands, calculated by comparing the relative area occupied by *Heterobasidion* sp. on treated and untreated halves of the disc surface, varied from negative results obtained in two spruce and one pine stand subjected to *P. gigantea* treatment to a complete control effect achieved in three urea- and one *P. gigantea*-treated pine stands (Table 2). The average efficacy of urea treatment was 99.5% on pine stumps and 85.3% on spruce stumps. The corresponding figures for *P. gigantea* treatment on pine and spruce stumps were 54.3% and 37.3%, respectively. Urea treatment prevented *Heterobasidion* infections more effectively on pine than spruce stumps (*p* < 0.001), while the efficacy of *P. gigantea* treatment did not differ significantly between the tree species (*p* = 0.924).

When all harvesting options were combined, the proportion of the cross-sectional area occupied by *Heterobasidion* sp. was significantly smaller in urea- than in *P. gigantea*-treated spruce stumps, i.e., 2.1 and 12.9%, respectively (t = 7.969, df = 175.273, *p* < 0.001). In pine stumps, the difference between the treatments was smaller, although statistically significant, i.e., 0.04 and 1.8% in urea- and *P. gigantea*-treated stumps, respectively (t = 4.377, df = 80.474, *p* < 0.001).

In general, the proportion of infected stumps and the relative area occupied by *Heterobasidion* sp. (E%) corresponded to each other, i.e., a high frequency of infected stumps indicated low control efficacy. However, the urea treatment in spruce stands was an exception because the average mean control efficacy was as high as 85.3%, even though 60.9% of treated stumps were infected by *Heterobasidion* sp. This was a consequence of the fact that the cross-sectional area occupied by *Heterobasidion* sp. was much smaller in urea-treated than *P. gigantea*-treated spruce stumps.

The thickness of the peat layer had no clear effect on the infected area on spruce stumps (Table 3). However, linear mixed models applied separately for spruce and pine stands showed variation in the treatment efficacy (based on the proportion of the cross-sectional area occupied by *Heterobasidion* sp.) between the harvesting methods. In spruce stands subjected to cap cutting, the relative area occupied by *Heterobasidion* sp. on stump surfaces (%) was lower in the stump sections treated with *P. gigantea* or urea than in the control sections (Table 3, Figure 1). In thinned spruce stands, instead, the proportion of infected areas was even higher in the *P. gigantea* treatment than in the control. Both in cap cutting and thinning, the infected area was low in the urea treatment.

In the clear-cut pine stands, the proportion of infected area was lower in the *P. gigantea* and urea treatments than in the control (Table 3, Figure 2). In the controls of cap cutting and thinning, the infected areas were larger than in the control of clearcutting. In the *P. gigantea* treatment of cap cutting, the proportion of infected area was larger than in clear-cutting and thinning. In the urea treatment, the infected area was low in all harvesting methods.

In *P. gigantea*-treated spruce thinning and pine cap cutting, the percentage of the area occupied by *Heterobasidion* sp. increased with increasing stump diameter. In the spruce stumps, the percentage of the occupied area increased from ca. 6 to 14% when the stump diameter increased from 5 to 40 cm, and in the pine stumps, from ca. 1 to 4% when the stump diameter increased from 5 to 35 cm. In other treatments, the control efficacy was good and was not affected by the stump diameter (Table 3 and Figure 3 and Figure 4).

Comparing the pooled data by tree species, the initial moisture content measured before the treatment was significantly higher in pine than in spruce sapwood, i.e., mean 116 and 100%, respectively (t = 4.422, df = 357.840, *p* < 0.001) while the moisture content of heartwood did not differ significantly between spruce and pine, i.e., 46 and 45%, respectively (t = −0.862, df = 359, *p* = 0.389). Detailed information on the stump moisture contents is given in Table 4.

On both pine and spruce stumps, the *Heterobasidion* colonies occurred almost exclusively in sapwood concentrating near the border zone between the sapwood and heartwood. The moisture content of sapwood and heartwood did not affect the area occupied by *Heterobasidion* sp. on the stump surface (Table 3).

## 4. Discussion

### 4.1. Heterobasidion Infections

The experimental areas of the present study are mainly located in the central part of Finland, where the prevalence of Heterobasidionroot rot is relatively low, and spore dispersal is more variable and forest-specific than in the most severely damaged areas in southern Finland [44]. Based on the infection of untreated control stumps, the natural deposition of *Heterobasidion* spores was absent on all other sites except Suonenjoki and Hämeenlinna. Light artificial inoculation to ensure *Heterobasidion* infections was therefore necessary. However, despite artificial inoculation, the pine clearcutting in Kuru had to be rejected because of the unsuccessful inoculation (only 3 out of 48 stumps became infected). The reason for the failed *Heterobasidion* infection was not determined but may have been due to the intensive natural colonization of pine stumps by *P. gigantea* [45]. Although the area colonized by *P. gigantea* was not exactly quantified, it was conspicuous that the sapwood of the untreated control stumps was almost completely occupied by the fungus. Actually, the untreated halves of most pine stumps were even more intensively colonized by natural *P. gigantea* than the halves treated with *P. gigantea*. Favorable moisture conditions in peatlands due to the high water-holding capacity of peat may improve the competitiveness of *P. gigantea* by increasing its sporulation. It is also known that there is a high variation in competitiveness among different *P. gigantea* strains [24], and an exceptionally aggressive wild strain able to displace *Heterobasidion* sp. on the stump surface may have been present in the cutting area in Kuru. Moreover, the competitiveness of *P. gigantea* may be improved by a low density of *Heterobasidion* inoculum [13,46]. In order to simulate the low level of the natural spore load in Central Finland [47], the inoculum concentration was considerably lower in the present study than in most of the earlier stump treatment studies, with a spore amount varying from 20 to thousands of spores per cm^−2^, e.g., [48,49,50]. Besides, the spore concentration in the inoculum used in Kuru was lower than that in the other experimental areas (2.1 vs. 4.7 viable spores/cm^−2^). Failure of *Heterobasidion* infection has also been reported in inoculation trials carried out on mineral soils [10,51,52], and therefore, it seems not to be connected to peatland conditions. Despite the exception in Kuru, the infection rate of untreated control sectors was rather high in spite of the low *Heterobasidion* spore load, which indicates that in boreal peatland forests, both spruce and pine stumps are prone to *Heterobasidion* spore infection. This observation is consistent with the earlier result obtained in Scotland, where no significant differences in the susceptibility of *Picea sitchensis* and *Pinus contorta* stumps to *Heterobasidion* infection were found between mineral and peat soils [48].

### 4.2. Efficacy of P. gigantea

There was a high variation in the efficacy of the biological control agent among the experimental stands. It failed in one pine stand and three spruce stands, while in two pine and spruce stands, *P. gigantea* treatment effectively prevented *Heterobasidion* infections (E% 89.7–100) (Table 2). In efficacy trials carried out in mineral soil, apart from a few exceptional cases, *P. gigantea* has provided sufficient protection against *Heterobasidion* infections with an efficacy (based on the relative area occupied by *Heterobasidion* sp.) ranging in spruce stumps from 70% to 100% [10,13,45,53] and from 86% to 100% in pine stumps [45]. In efficacy trials carried out on mineral soils, a poor result of *P. gigantea* treatment has been connected to incomplete coverage of the control agent on the stump surface [26,27] and exceptionally high *Heterobasidion* spore pressure [13,54]. Neither of these factors can be applied to our material because the control agent was sprayed manually, ensuring complete coverage of the stump surface, and the density of *Heterobasidion* spores in the inoculum was lower than usually used in efficiency trials carried out on mineral soils.

It should be noted that all the *P. gigantea* treatments providing poor control results in our study were established in 2021. Therefore, the harvesting method itself, i.e., cap cutting in a pine stand and thinning in some spruce stands, does not necessarily explain the substandard result. A different production batch of Rotstop^®^ SC was used each year. The quality (spore viability, growth rate on agar, and purity) of Rotstop^®^ SC is tested by the manufacturer [55], and the product was stored and handled according to the instructions of the manufacturer, and therefore the viability of oidia was not determined during the present study. Hence, we cannot rule out the possibility that there may have been differences in control efficacy between the batches used.

The efficacy of a biological control agent may also be affected by environmental factors. All treatments in 2021 were carried out during a short period between the end of July and the end of August. The early summer (June–July) of 2021 was exceptionally dry and warm, with temperatures 3.0–4.6 degrees above long-term averages, while at the end of summer, there were locally heavy rain showers [56]. Because of the absence of experiment-specific weather information, no conclusions can be made about the possible influence of precipitation on the control efficacy. Both rainfall and wetness of the soil affect the moisture content of stump wood which is an important factor affecting the susceptibility of spruce and pine stumps to fungal spore infections regarding both pathogenic and biocontrol fungi [29,57]. In our material, the initial moisture content of pine sapwood was significantly higher in experiments established in 2021 than in 2020, i.e., 133% and 107%, respectively, while among spruce stumps, no statistically significant differences between the years were obtained. Although the proportion of area colonized by *Heterobasidion* sp. decreased with increasing stump moisture content in Norway spruce stumps in a Swedish study [58], our results did not show—despite the annual moisture variation in pine stumps—any relationship between the initial moisture content of sapwood or heartwood and the proportion of area colonized by *Heterobasidion* sp. on treated stump sectors.

### 4.3. Efficacy of Urea

Urea showed better control efficacy than *P. gigantea* both on spruce and pine stumps. Although in spruce stands (pooled harvesting method data), the proportion of infected stumps was high both in urea and *P. gigantea* treatments, i.e., 60.9 and 70.2%, respectively, the efficacy of urea was nevertheless high (mean 79.5%) due to the small relative area occupied by *Heterobasidion* sp. on stump surfaces. Only in the thinned spruce stand in Multia, established in 2021, the control result of urea was substandard (E% 42). Overall, there seems not to be significant differences in the control efficacy in spruce stumps between peat and mineral soils, as the efficacy of 30% urea has previously been shown to vary between 60–100% [10,12]. In our pine stands, urea treatment proved to be very effective, i.e., E% 99.6–100% (mean 99.9%), and even higher than that obtained on pine stumps on mineral soils [17,22,59,60].

Because drying out of the stump surface reduces the efficacy of urea treatment, a dry period in summer may hinder urea hydrolysis leading to reduced control efficacy [11]. However, we did not find any effect of dry summer (in 2021) on stump moisture content. Moreover, in the spruce stands in Suonenjoki, where the urea treatment was conducted in 2021, the control efficacy was high.

On the other hand, urea treatment may fail due to the excessive moisture of stump wood. In efficacy trials carried out in Scotland and northern England, a urea treatment of Sitka spruce stumps did not protect stumps against artificial *H. annosum* inoculum. Particularly on peat soil, urea treatment promoted the colonization of *H. annosum* and increased the cross-sectional area colonized by *H. annosum* [28]. The authors suggested that *Heterobasidion* benefits from additional nitrogen, especially in heartwood which is lower in nitrogen than sapwood. Because heartwood is dryer than sapwood, it is preferred by the fungus on a peatland subjected to high annual rainfall. In stumps analyzed during the present study, *Heterobasidion* infections were concentrated on the sapwood, and the boundary between sapwood and heartwood, similarly as in stumps on mineral soil [58], and no promoting effect of urea treatment on *Heterobasidion* infection was observed. Only in 4% of the urea-treated spruce stumps, the relative area occupied by *Heterobasidion* sp. was greater on treated than untreated half of the stumps.

The efficiency of urea also depends on the concentration of the treatment solution. A more concentrated urea solution (>30%) provides better protection than a more diluted solution [12,14]. Thus, in our study, one reason for the better control efficacy of urea treatment compared to an earlier result [28] may be that we used a 30.5% urea solution instead of the more diluted 17% solution used in the earlier experiment.

### 4.4. Effects of Stump and Soil Characteristics on the Control Efficacy

#### 4.4.1. Stump Moisture Content

Unexpectedly, the moisture content of spruce stumps on peatland was, on average lower than those reported in mineral soil. In mineral soil, an average moisture content of 148% has been reported by Bendz-Hellgren and Stenlid [58] for sapwood of Norway spruce thinning stumps, while the corresponding moisture content in our thinning material was 97%. Moreover, in all our spruce cap-cutting treatments, the moisture content of sapwood was lower (mean 114%) than in clearcutting stumps in mineral soil (163%) [58]. Moreover, Shain [61] reported higher wood moisture contents of sapwood of Norway spruce trees in mineral soils (127.4%) than what we observed in our peatland stands (107%). In Scots pine stumps, instead, lower moisture contents varying from 80 to 90% have been obtained in mineral soil [29] than in peat soil in the present study (112%). In all, our results reveal that, in boreal forests, there seems not to be a marked difference in the stump moisture contents between drained peatlands and mineral soils. Since no exceptional moisture contents occurred on peat soil, it is obvious that stump moisture content is not a critical factor affecting control efficiency either.

#### 4.4.2. Stump Diameter

Several studies have shown that the susceptibility of both spruce and pine stumps to *Heterobasidion* spore infection increases with increasing stump diameter [62,63,64,65]. At the same time, the best efficacy of urea treatment is expected in small-diameter stumps in young stands because urease activity is at its highest in young sapwood, while in dead heartwood, the activity is low [11]. Meanwhile, *P. gigantea* treatment gives better protection against *Heterobasidion* infection in large than small stumps [25].

In our material, the reduced efficacy of *P. gigantea* treatment with increasing stump diameter was evident in the pine cap cutting and spruce thinnings (Figure 3 and Figure 4). The stands were, however, rather old (74–110 years), and small-diameter trees were generally not younger than larger ones but slow-growing understory trees with very narrow annual rings. In those slow-growing, old trees, the percentage of heartwood is greater than in dominant trees of the same age [66]. In this light, stump treatment seems to provide good protection not only for small stumps of young trees but also for small stumps of slow-growing undergrowth trees.

#### 4.4.3. Soil pH

In terms of the soil pH, *Heterobasidion* species can adapt to a wide range of different habitats. On mineral soils, extensive damage of root rot caused by *H. annosum* has been reported in pine stands with a pH greater than 6 [67]. In turn, Gaitnieks et al. [68] reported on a Norway spruce stand suffering from Heterobasidionroot rot on peat soil with a pH as low as 2.6. In our study, the soil pH values were quite low (3.7–4.0) without much variability between the experimental stands, and no direct effect of pH on stump infections or control efficacy was found. However, the effect of the low soil pH might appear later by slowing down the mycelial spread of the fungus in the root systems.

#### 4.4.4. Thickness of the Peat Layer

According to our knowledge, no direct information is currently available on the possible connection between the thicknesses of the peat layer and the risk of primary *Heterobasidion* infection. The data based on the national forest inventory of southern Finland revealed, however, that a peat layer thicker than 30 cm decreases the risk of butt rot on Norway spruce (mainly caused by *H. parviporum*) [69]. In our material, the thickness of the peat layer had no significant effect on the susceptibility of stumps to *Heterobasidion* infection or on the control efficacy (proportion of area occupied by *Heterobasidion* sp.), indicating that stump treatment may be advisable also on peat soils with a peat layer thicker than 30 cm. Although peatland forests seem not to be an optimal habitat for *Heterobasidion* sp., some recent studies have demonstrated that Heterobasidionroot rot can cause severe damage for both spruce and pine in peatlands [38,70]. To save peatland forests from Heterobasidionroot rot, efficient prophylactic control measures would be important. Not only performing stump treatment but also avoiding logging damage in peatland forests, where, due to the subsidence of the peat surface after drainage, the risk of mechanical injuries to the roots and stem base increases [37].

To conclude, the present study reveals that both spruce and pine stumps are susceptible to *Heterobasidion* spore infection on drained peatlands, which encourages the use of stump treatment in summer cuttings. Urea as a 32.5 percent aqueous solution gave good protection against *Heterobasidion* infections both in pine and spruce stumps, while *P. gigantea*’s efficacy varied greatly among experimental stands. Further research is needed to determine the reason that led to the failure of the biological control agent, notwithstanding the control agent covered the entire surface of the stumps and the pressure of *Heterobasidion* spores was extremely low. Since this study provides information on the short-term efficacy of the stump treatments after harvesting, it is important to evaluate the efficacy over a longer period as well. In addition, the possible environmental effects of urea on the peatland ecosystem should be investigated.

## Figures and Tables

**Figure 1 jof-09-00346-f001:**
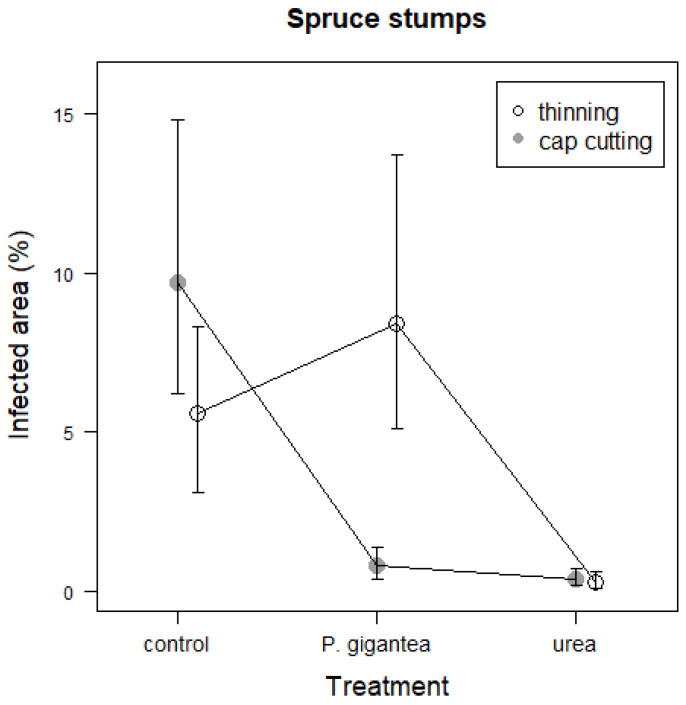
Area covered by *Heterobasidion* sp. (%) in the control, *P. gigantea*, and urea treatments in Norway spruce stumps in thinning and cap cutting. Predicted values with standard errors of means (SE) in different treatments and harvesting methods based on the estimated model are presented.

**Figure 2 jof-09-00346-f002:**
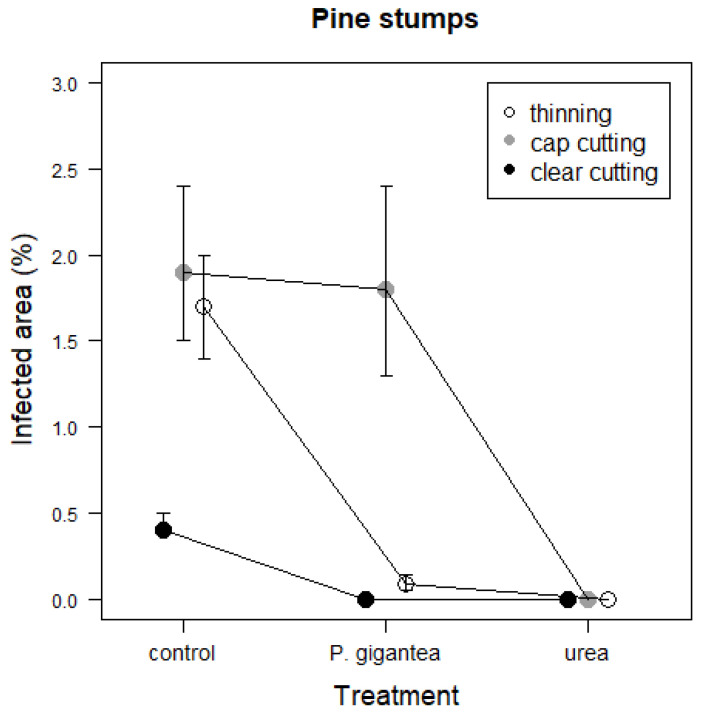
Area covered by *Heterobasidion* sp. (%) in the control, *P. gigantea*, and urea treatments in Scots pine stumps in thinning, cap cutting, and clearcutting. Predicted values with standard errors of means (SE) in different treatments and harvesting methods based on the estimated model are presented.

**Figure 3 jof-09-00346-f003:**
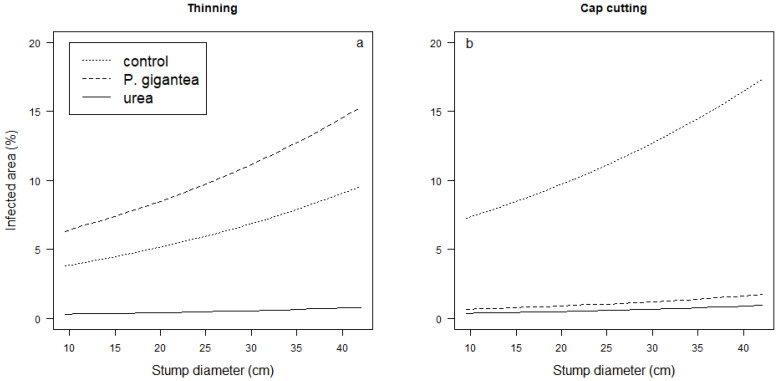
Area covered by *Heterobasidion* sp. (%) in spruce stumps as a function of stump diameter in different treatments (control, *P. gigantea*, or urea) after (**a**) thinning and (**b**) cap cutting. Predicted values based on linear mixed models have been used to draw the figure.

**Figure 4 jof-09-00346-f004:**
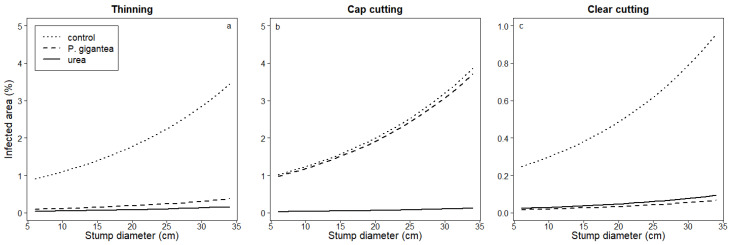
Area covered by *Heterobasidion* sp. (%) in pine stumps as a function of stump diameter in different treatments (control, *P. gigantea*, or urea) after (**a**) thinning, (**b**) cap cutting, and (**c**) clear-cutting. Predicted values based on linear mixed models have been used to draw the figure.

**Table 1 jof-09-00346-t001:** Description of experimental stands.

Experimental Stand	Location (Lat/Lon)	Site Type ^a^	Soil pH	Avg. Thickness of the Peat Layer, cm	Tree Species	Avg. Stand Age, Years	Avg. Stump Diameter ^b^ cm	Type of Cutting
Kuru	61°59′37.109″	MtkgII/	3.8	135	pine	65	21	clearcutting
	23°41′13.743″	Ptkg II						
Multia 1	62°26′55.351″	Ptkg II	4.0	150	pine	117	18	thinning
	25°0′28.664″							
Keuruu 1	62°29′27.446″	Ptkg II	3.8	110	pine	78	21	clearcutting
	24°21′14.783″							
Keuruu 2	62°29′35.835″	Ptkg II	3.7	90	pine	65	12	thinning
	24°21′15.578″							
Suonenjoki 1	62°33′40.972″	MtkgII	3.8	60	pine	110	25	cap cutting
	27°14′59.115″							
Suonenjoki 2	62°33′46.092″	Rhtkg I	3.8	50	spruce	103	24	cap cutting
	27°14′50.895″							
Suonenjoki 3	62°33′35.468″	Rhtkg I	3.9	50	spruce	98	21	thinning
	27°15′5.650″							
Multia 2	62°32′14.467″	Mtkg I/	3.9	80	spruce	74	17	thinning
	24°57′55.717″	Ptkg I						
Hämeenlinna	61°11′5.974″	Mtkg II	3.9	60	spruce	88	22	cap cutting
	25°15′46.907″							
Vesijako	61°25′11.925″	Mtkg I	3.9	80	spruce	78	18	cap cutting
	25°0′23.800″							

a Herb-rich type (Rhtkg), *Vaccinium myrtillus* type (Mtkg), and *Vaccinium vitis-idaea* type (Ptkg). Subtype I: originally forest-covered mires, subtype II: originally treeless or mixed mires. b Without bark.

**Table 2 jof-09-00346-t002:** Treatment date and duration of the experiment; inoculum density (number of viable *Heterobasidion* spores per cm^−2^); the number of treated, analyzed, and infected stumps; and treatment efficacy in experimental stands.

Experimental Stand	Treatment Date	Duration of the Exp., Day	No. of *Het* Spores, per cm^−2^	No. of Treated/Analyzed Stumps		No. of Stumps inf. by *Heterobasidion* (%)		Treatment Efficiency, %	
				Urea	*P. gigantea*	Urea	*P. gigantea*	Urea	*P. gigantea*
*Pine stands*									
Kuru	28.7.2020	62/76 ^a^	2.1	25/1	29/2	-	-	-	-
Multia 1	25.8.2020	63	NA	21/13	25/21	1 (7.7)	6 (28.6)	99.6	89.7
Keuruu 1	26.8.2020	75	3.0	25/11	25/14	0	0	100	100
Keuruu 2	26.8.2020	83	3.0	24/17	25/18	0	6 (33.3)	100	22.9
Suonenjoki 1	11.8.2021	69	4.6	30/30	30/27	0	26 (96.3)	100	−56.6
*Spruce stands*									
Suonenjoki 2	30.7.2021	66	4.4	28/27	30/30	21 (77.8)	30 (100)	87.1	−29.3
Suonenjoki 3	30.7.2021	67	4.4	25/23	30/30	14 (60.9)	30 (100)	92.0	5.9
Multia 2	23.8.2021	65	7.2	30/20	30/25	16 (80.0)	23 (92.0)	41.9	−39.4
Hämeenlinna	20.6.2022	78	4.7	30/28	28/27	18 (64.3)	7 (25.9)	79.5	98.2
Vesijako	21.6.2022	76	3.0	30/28	30/27	6 (21.4)	9 (33.3)	97.2	95.9

^a^ Stump samples were collected on two occasions.

**Table 3 jof-09-00346-t003:** The effects of treatment (*P. gigantea* or urea), harvesting method (clearcutting, cap cutting or thinning), stump diameter (cm), wood moisture (%), and peat thickness (cm) on the infected area by *Heterobasidion* (%) on cut stumps. Coefficients and standard errors of means (SE) have been presented for linear mixed models. Negative effects on the response, i.e., the infected area by *Heterobasidion* (%), are indicated as negative and positive ones as positive coefficients. Models are presented separately for Norway spruce (*n* = 535) and Scots pine stump surfaces (*n* = 351). Values in the table are in logit scale since the response was logit transformed when the models were estimated. *p*-values < 0.05 are in bold.

Explanatory Variables	Spruce Stump Model		Pine Stump Model	
	Coeff. ± SE	*p*	Coeff. ± SE	*p*
Intercept	0.666 ± 0.884	0.452	−6.139 ± 0.753	<0.001
*P. gigantea* ^a^	−2.489 ± 0.199	<0.001	−2.667 ± 0.360	<0.001
Urea ^a^	−3.106 ± 0.201	<0.001	−2.325 ± 0.404	<0.001
Cap cutting ^b^	-	-	1.427 ± 0.337	0.148
Thinning ^b^	−0.682 ± 0.315	0.275	1.307 ± 0.299	0.143
Stump diameter ^c^	0.030 ± 0.014	0.035	0.048 ± 0.020	0.014
Moisture in sapwood ^d^	0.002 ± 0.002	0.251	−0.328 × 10^−3^ ± 0.004	0.934
Moisture in heartwood ^d^	−0.006 ± 0.005	0.235	−0.003 ± 0.008	0.741
Thickness of peat ^e^	−0.054 ± 0.009	0.102	-	-
*P. gigantea*: cap cutting ^f^	-	-	2.625 ± 0.448	<0.001
Urea: cap cutting ^f^	-	-	−1.106 ± 0.476	0.022
*P. gigantea*: thinning ^f^	3.022 ± 0.320	<0.001	0.417 ± 0.423	0.326
Urea: thinning ^f^	0.510 ± 0.339	0.133	−0.735 ± 0.473	0.122

a, b In spruce stumps, the difference is compared to the control in cap cutting. In pine stumps, the difference is compared to the control in clearcutting. c Without bark (cm). d Measured before the treatments (%). e Peat thickness has been measured as cm. f. Interaction between the treatments (control, *P. gigantea*, or urea) and the harvesting methods (clearcutting, cap cutting, or thinning). These indicate differences when moving from the control to the other treatments (*P. gigantea* or urea) in the different harvesting methods.

**Table 4 jof-09-00346-t004:** Initial and final moisture contents (MC%) of sapwood and heartwood are given as a mean (range in parenthesis) in experimental stands.

Experimental Stand	Initial MC%	Final MC%	Initial MC%	Final MC%
	Sapwood	Sapwood	Heartwood	Heartwood
*Pine stands*				
Kuru	105 (42–144)	70 (40–169)	42 (32–59)	49 (32–74)
Multia 1	103 (57–149)	127 (68–174)	44 (32–84)	46 (31–121)
Keuruu 1	104 (42–249)	125 (84–170)	43 (35–68)	47 (33–75)
Keuruu 2	115 (56–181)	135 (74–181)	44 (15–60)	54 (31–103)
Suonenjoki 1	133 (58–270)	132 (71–287)	48 (33–155)	44 (31–80)
*Spruce stands*				
Suonenjoki 2	120 (49–203)	134 (26–233)	53 (38–111)	61 (37–125)
Suonenjoki 3	79 (40–220)	90 (38–190)	49 (39–85)	51 (35–80)
Multia 2	114 (36–186)	160 (51–255)	40 (27–61)	56 (41–159)
Hämeenlinna	132 (30–200)	142 (33–278)	48 (32–197)	45 (22–142)
Vesijako	89 (32–207)	109 (40–195)	42 (33–59)	44 (33–74)

## Data Availability

All data presented in the manuscript are available upon request to the corresponding author.

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
