# Peer review of "Efficacy of Biological and Chemical Control Agents against Heterobasidion Spore Infections of Norway Spruce and Scots Pine Stumps on Drained Peatland"

_jof, 2023, doi:10.3390/jof9030346_

Round 1

Reviewer 1 Report

This paper undertakes a very important problem of protection methods and their efficacy against the most important tree disease in the boreal zone - Heteroabsidion root rot. In addition, the study was conducted in a specific environment - drained peatland stands. In many counties, there is a question about biological control methods in particular their effectiveness usage, which could be the threshold of real profitability. Questions about sometimes low effectiveness of the treatment despite the common use. Various factors affecting the success of the treatment are considered. Part of this study addresses these questions. The authors take into account many factors affecting both the effectiveness of the procedure and the infection caused by Heterobasidion

This paper provides new information about different agents influencing Phlebiopsis colonization of treated stumps. I must emphasize that the design of the experiment, the quality of the presentation of the results, and the discussion are excellent. In addition, the wide scope of research, and numerous research areas, but also a huge amount of empirical data increase the importance of this work. This manuscript points out the urea I concentration > 30% as a good agent for stump protection against Heterobnasidion spore infection but also characterizes the limitation of usage.

I recommend publishing this manuscript without revision.

Author Response

Reviewer #1

We thank for the positive appraisal of our manuscript.

Reviewer 2 Report

Thank you for the interesting manuscript.

Heterobasidion annosum s.l. is a threat to conifer cultivation, especially in northern europe. The study investigates two prophylactic control options with Urea and P. gigantea. The novelty of the study is that the experimental set-up takes place in boreal drained peatland.

The text is written in an understandable way, the experimental set-up is clear and the results are well presented.

I have only added some small suggestions for improvement and a few comments and questions to the PDF.

Author Response

Responses to comments (Reviewer #2)

  1. I think here the two species could be mentioned.

Both Heterobasidion species occurring in Finland, H. parviporum and H. annosum, are now mentioned in the text.

  1. in line 151 a slightly different name is mentioned

“PS-kantosuoja” has been changed to “PS-kantosuoja-2”.

  1. this section is very long and i don't know if everything is relevant (vegetation types, subtypes..) please check and shorten if necessary

The section has been shortened as suggested.

  1. in all tables a different font is used than in the text, please adapt it

The font in Tables has been changed.

  1. why is the species name not italicized throughout the manuscript?

In the submitted version of the manuscript the scientific names were in italics; we have changed them to italics again.

  1. how was this done?

The method has been added to the text.

  1. i think here it would have been good to test the germination rate in vitro. maybe you can also discuss this (different batches of rotstop)

The germination of the P. gigantea should definitely have been checked, but we trusted the information received from the manufacturer and did not do it – unfortunately. See also Discussion; Lines 426-431.

  1. italics?

See the answer to comment 5.

  1. is there a reason? i know time and money are always limited

Yes, we have microscoped and measured the area occupied by Heterobasidion sp. in more than 500 stumps and doing the same work regarding the presence of P. gigantea would have been too much work with the available resources. Thus, we had to accept a less accurate, visual assessment. We should have been noted this extra work in the research plan.

  1. important :-)

Yes.

  1. italics

See the answer to comment 5.

  1. delete

Deleted.

  1. sometimes there are blanks between equals and number. please check!

line 321

The blank spaces have been inserted through the text.

  1. would it be feasible to produce a self-made solution from autochthonous aggressive strains?

It could be possible but, unfortunately, very difficult to implement in practice.

  1. italics

See the answer to comment 5.

  1. could the impact of urea on the fragile boreal ecosystem also be studied?

This is a good point. We added the sentence: “In addition, the possible environmental effects of urea on the peatland ecosystem should be investigated.”

The comments made by the editor (I-VII) have been taken into account in the revised text.